# A Possible Link between Cell Plasticity and Renin Expression in the Collecting Duct: A Narrative Review

**DOI:** 10.3390/ijms25179549

**Published:** 2024-09-03

**Authors:** Nicole Schary, Bayram Edemir, Vladimir T. Todorov

**Affiliations:** 1Department of Physiology and Pathophysiology, Center of Biomedical Education and Research (ZBAF), Faculty of Health—School of Medicine, Witten/Herdecke University, 58453 Witten, Germany; nicole.schary@uni-wh.de; 2Department of Internal Medicine IV, Hematology and Oncology, Martin Luther University Halle-Wittenberg, 06120 Halle, Germany; 3Experimental Nephrology and Division of Nephrology, Department of Internal Medicine III, University Hospital and Medical Faculty Carl Gustav Carus, Technische Universität Dresden, 01307 Dresden, Germany

**Keywords:** collecting duct function, cell differentiation, cell plasticity, renin–angiotensin–aldosterone system (RAAS), nephrogenesis, signaling, gene expression

## Abstract

The hormone renin is produced in the kidney by the juxtaglomerular cells. It is the rate-limiting factor in the circulating renin–angiotensin–aldosterone system (RAAS), which contributes to electrolyte, water, and blood pressure homeostasis. In the kidneys, the distal tubule and the collecting duct are the key target segments for RAAS. The collecting duct is important for urine production and also for salt, water, and acid–base homeostasis. The critical functional role of the collecting duct is mediated by the principal and the intercalated cells and is regulated by different hormones like aldosterone and vasopressin. The collecting duct is not only a target for hormones but also a place of hormone production. It is accepted that renin is produced in the collecting duct at a low level. Several studies have described that the cells in the collecting duct exhibit plasticity properties because the ratio of principal to intercalated cells can change under specific circumstances. This narrative review focuses on two aspects of the collecting duct that remain somehow aside from mainstream research, namely the cell plasticity and the renin expression. We discuss the link between these collecting duct features, which we see as a promising area for future research given recent findings.

## 1. Introduction

The renal collecting duct is an important segment of the nephron involved in the final concentration or dilution of urine, a central process in maintaining proper fluid balance in the body. The collecting duct plays a pivotal role in the regulation of water and electrolyte balance, acid–base homeostasis, and blood pressure control [1]. To ensure the proper balance of water and electrolytes in the body, hormones such as aldosterone, antidiuretic hormone (ADH), and atrial natriuretic peptide [2] act on the collecting duct to modulate its function in response to changes in the body’s hydration status and electrolyte levels.

The collecting duct exhibits cellular plasticity that allows it to adapt to different physiological and environmental conditions. The cellular plasticity of the collecting duct refers to the ability of its cells to undergo functional and structural changes and thus switch between different phenotypes. For example, principal cells, which are primarily responsible for sodium and water reabsorption, can differentiate into intercalating cells, which are important for maintaining acid–base balance and vice versa. While the transition between these cell types is rare in healthy tissues, certain conditions and mutations can alter this balance [3].

Interestingly, cellular plasticity is a hallmark feature of the juxtaglomerular (JG) cells in the renal afferent arterioles. These cells produce renin that controls the activity of the circulating renin–angiotensin–aldosterone system (RAAS), which in turn is a key player in the regulation of blood pressure and renal function [4,5,6,7]. Furthermore, renin expression was observed in the collecting duct cells [8]. Renin is also regarded as a plasticity marker of the renin-producing cells in the renal vasculature (for recent reviews, see refs [4,5,6]). Moreover, renin expression and cellular plasticity in the collecting duct are documented. Therefore, it is conceivable to assume that a link between these phenomena in the collecting duct also exists. Our narrative review addresses the current evidence and possible future directions in the context of this new concept.

## 2. Functions of the Renal Collecting Duct Cell Types

The collecting duct can be categorized into the cortical collecting duct, the outer medullary collecting duct, and the inner medullary collecting duct based on its location within the kidney [9]. Along the collecting duct, three main epithelial cell types are present, each with specialized functions.

Principal cells are primarily responsible for maintaining the body’s fluid and electrolyte balance. With this regard, the main membrane channels in principal cells are aquaporin-2 (AQP2), the epithelial sodium channel (ENaC), and the renal outer medullary potassium (K) channel (ROMK) on the apical membrane [10], which collectively result in fine-tuning of the reabsorption of sodium and water from the tubular fluid and the secretion of potassium into the collecting duct lumen.

The expression of the water channel AQP2 is tightly regulated by ADH [11]. Under basal conditions, AQP2 is found predominantly within storage vesicles located below the apical cell membrane. The translocation of AQP2 from storage vesicles to the apical membrane is induced upon binding of ADH to the V_2_ receptor on the basolateral side of the principal cells [12,13]. ADH is released from the posterior pituitary in response to changes in the extracellular fluid osmolarity [14]. The binding of ADH to the V_2_ receptor increases intracellular cAMP levels, which in turn activates protein kinase A (PKA). Thereupon, PKA phosphorylates AQP2 and triggers its translocation into the apical membrane [15], resulting in increased water permeability.

Another channel expressed in principal cells is ENaC, a heterotrimeric ion channel composed of α, β, and γ-ENaC subunits [16,17,18]. This channel facilitates the apical entry of sodium ions into principal cells and is crucial for maintaining sodium balance and blood pressure homeostasis. A central role in the regulation of ENaC is the corticosteroid aldosterone, which influences its expression and activity. On the one hand, aldosterone increases the expression of the α, β, and γ-subunits of ENaC [19,20], a process mediated by the mineralocorticoid receptor (MR). On the other hand, aldosterone promotes the transcription of serum- and glucocorticoid-induced kinase 1 (SGK1), which subsequently activates ENaC through various mechanisms [21,22,23]. Electrogenic sodium transport via ENaC increases the driving force for potassium secretion [24] via ROMK, expressed in the luminal membrane of the principal cells [25,26]. Expression and activity of ROMK are influenced by various factors, which could be subdivided into aldosterone-dependent and aldosterone-independent and include high potassium intake, PKA phosphorylation, intracellular pH, or with-no-lysine (K) (WNK) kinases [27,28,29]. Thus, the interdependence of ENaC and ROMK is essential for the body’s electrolyte balance, as their coordinated function not only regulates sodium entry and potassium secretion but also ensures overall electrolyte homeostasis in response to hormonal and dietary influences [30].

In addition to their important role in the body’s fluid and electrolyte homeostasis, the channels AQP2, ENaC, and ROMK are also linked by angiotensin II (Ang II), which is part of the RAAS (see also Section 3.2 below).

Li et al. demonstrated that Ang II regulates the AQP2 gene and protein expression in principal cells in vitro. This Ang II effect could be blocked by the AT1 receptor inhibitor losartan. This study also showed that treatment with Ang II activates the cAMP, PKA, PKC, and calmodulin signaling pathways [31]. Likewise, treatment of rats with the AT1 receptor inhibitor candesartan reduced ADH-induced water reabsorption and lowered AQP2 concentrations under NaCl restriction in vivo [32]. With regard to ENaC, several studies have described that Ang II can influence its posttranscriptional modifications [33] and its opening probability [34,35,36]. In the case of ROMK, Ang II inhibits ROMK by decreasing c-Src activation and blocking ROMK internalization [37].

Furthermore, intercalated cells are primarily responsible for maintaining acid–base balance. There are two subtypes of intercalated cells: type A intercalated cells (or α-intercalated cells) and type B intercalated cells (or β-intercalated cells).

Specifically, α-intercalated cells utilize H^+^-ATPase (comprising ATP6V0A1-4 and ATP6V1B1-2 subunits [38,39]) and H^+^/K^+^-ATPase/ATP4A at the apical membrane to secrete hydrogen into the collecting duct lumen [40]. In turn, the sodium–hydrogen antiporter 1 (NHE1)/SLC9A1 is expressed basolaterally and transports hydrogen into the blood, thus balancing the overall buffering capacity of the blood and preventing acidosis. Additionally, the absorption of potassium is mediated through H+/K+-ATPase/ATP12A located in the apical membrane, allowing the uptake of potassium ions in exchange for hydrogen ions. The excretion of potassium is further facilitated by a big potassium (BK) channel/KCNMA1. Bicarbonate can be moved into the blood via basolateral anion exchanger 1 (AE1)/SLC4A1. (reviewed elsewhere [30,41]).

On the other hand, β-intercalated cells apically express pendrin/SLC26A4 to secrete bicarbonate into urine and reabsorb chloride, and sodium-driven chloride/bicarbonate exchanger (NDCBE)/SLC4A8 to reabsorb bicarbonate and sodium and secrete chloride. On the basolateral side, bicarbonate in β-intercalated cells is transported via the Na^+^-dependent HCO3^−^-exchanger AE4/SLC4A9 [42]. Furthermore, hydrogen is transported via H^+^-ATPase, which operates with the participation of subunits ATP6V0A1-4 and ATP6V1B1-2 [38,39]. By equipping α-intercalated cells and β-intercalated cells with these proteins, these cells play a crucial role in maintaining the pH of the blood plasma within a narrow range [41].

Similar to principal cells, transporters of intercalated cells can also be influenced by components of the RAAS. Ang II promotes the expression of pendrin [43,44], whereby an interaction between aldosterone and Ang II for the upregulation of pendrin is also discussed [45]. It has also been demonstrated that Ang II enhances chloride absorption via pendrin; however, Ang II does not exert direct effects on pendrin but rather influences the activity of H+-ATPase [46].

Altogether, these findings underline a substantial interplay between the principal cells/the intercalated cells and the RAAS, indicating that the collecting duct may be intricately modulated by the regulatory mechanisms of the RAAS.

The transcellular transport processes in the principal cells and intercalated cells are decisive for the function of the collecting duct. However, the expression of claudins, a group of pore-forming proteins, at the epithelial cell–cell contacts can facilitate the paracellular transport of chloride and/or sodium [47]. Together with the intercalated cells, the expression pattern of claudins mediates the chloride transport [47]. It is currently unclear to what extent the paracellular pathway contributes to the overall transepithelial transport in the collecting duct.

## 3. Cellular Plasticity

Considering the compatible functions of the cell types that compose the collecting duct, it is not surprising that they are capable of differentiating from each other to adapt renal function to various environmental cues. Such a feature is generally referred to as cellular plasticity. From a physiological standpoint, cellular plasticity could be defined as a shift in the functional phenotype accompanied by structural changes in a cell.

Cellular plasticity has been first defined as a non-neoplastic change in cell identity in pathology [48]. Cancer phenotype transformation is also an example of pathological cell plasticity. Cellular plasticity is further attributed to developmental events describing the differentiation of specialized progeny cells derived from stem cells. This phenomenon also persists during adult life in organs and tissues with high cell turnover rates, such as bone marrow or intestinal epithelia. The increasing number of processes related to phenotype alterations that fall under the term cellular plasticity made the development of a nomenclature necessary [49]. The definitions proposed are still discussed because they may address complex sequels with completely different outcomes. For instance, epithelial-mesenchymal transition (EMT) is an event featuring cell plasticity observed in embryonic development and regeneration after injury, but also during organ fibrosis or malignant transformation [50].

### 3.1. Plasticity in the Renal Collecting Duct

Cellular plasticity was also observed in the collecting duct using different experimental approaches. For example, in in vitro experiments mimicking metabolic acidosis, β-intercalated cells differentiated into α-intercalated cells when cultured under acidic conditions [51,52]. While earlier studies have shown that β-intercalated cells can acquire α-intercalated cells and even principal cell-like properties, differentiation from principal cells to intercalated cells was not observed [53]. The conversion of β-intercalated cells into α-intercalated cells was abolished in mice lacking extracellular matrix protein hensin. These mice showed no signs of α-intercalated cells. Thus, the deletion of hensin resulted in the development of distal renal tubular acidosis [54].

In carbonic anhydrase type II (CAII) deficient mice, the number of intercalated cells was reduced, and that of principal cells increased in the medullary collecting duct [55]. It could be postulated that CAII activity is necessary for the differentiation of principal cells of the medullary intercalated cells during development.

The fact that intercalated cells can originate from principal cells has been demonstrated by principal cell-specific deletion of the histone H3 K79 methyltransferase Dot1l [56]. Transgenic mice carrying this mutation had fewer principal cells and more intercalated cells. These data show that the plasticity in the collecting duct can change cell composition and ratio.

Progenitor cells have the capacity for self-renewal and clonogenicity and could form multiple cell types. In mice, AQP2- and V-ATPase subunit B1/B2-positive progenitor cells were capable of forming distal convoluted tubules (DCTs), connecting tubules (CNTs), and collecting duct segments during development [57]. The same progenitor cells were also identified in adult mice. These cells were involved in the maintenance and repair of the segments described above. However, Park et al., who identified stable transitional (or hybrid) cells between principal cells and intercalated cells, also suggested that the transition rate between the two cell types is low in healthy tissues [58].

Research in animal models demonstrated that Notch signaling is essential for maintaining the cellular composition of the collecting duct. Notch1, Notch2, and Hes1, a downstream target of the Notch signaling pathway [59], are essential for normal levels of AQP2 gene expression in principal cells. Inactivation of Notch1, Notch2, and Hes1 led to the transdifferentiation of mature principal cells into intercalated cells in the kidneys of adult mice [60,61]. Potassium depletion in rats increased the α-intercalated and decreased the principal cell numbers via suppression of Notch signaling through the Hes1 pathway [62]. Foxi1, which is suppressed by Hes1 [61], serves as a crucial transcription factor involved in the differentiation of intercalated cells because principal cell markers are selectively upregulated in Foxi1-deficient mice [63,64]. Importantly, deteriorated Notch signaling results in renal distal tubular disorders like nephrogenic diabetes insipidus (NDI) [55].

In summary, cellular plasticity within the collecting duct is a dynamic and multifaceted process that is influenced by various genetic and homeostatic variables. The capacity for transdifferentiation is modulated by factors like hensin, CAII, and Notch signaling. The current findings suggest that cellular plasticity serves as a compensatory mechanism crucial for maintaining the body’s homeostasis by regulating water, electrolyte, and pH balance.

### 3.2. Plasticity of the Renin-Producing Juxtaglomerular Cells and Renin Expression in the Collecting Duct

Next to the collecting duct cells, cellular plasticity as a physiological event in the mature kidney is also well established for the renin-producing cells in the terminal afferent arterioles. These cells are called juxtaglomerular (JG) because of their location in the glomerular vicinity. Renin is the rate-limiting enzyme in the generation of Ang II [7]. It serves as the major switch of the plasma RAAS that controls blood pressure as well as salt and water homeostasis. For this reason, the renin production in the afferent arterioles is tightly regulated to provide an adequate adaptation of the organism to environmental changes. The central factor that controls the renin synthesis and secretion in the JG cells is arterial blood pressure. Blood pressure decrease stimulates renin production, while blood pressure increase generates an opposite response [65,66]. In turn, the changes in renin release and, consequently, in RAAS activity induce Ang II-mediated effects, countersteering the initial alterations in arterial tone and resulting in blood pressure normalization. This pressure-dependent mechanism that regulates renin production in the afferent arterioles is known as the “long negative feedback loop” of RAAS [7,67]. The baroreceptor sensing the pressure fluctuations has long been proposed to reside in the renal vasculature because the long negative feedback loop is functional in the isolated perfused kidney [7,68]. Recently, the baroreceptor mechanism was identified in the JG cells themselves. It involves nuclear mechanotransduction, wherein extracellular forces (i.e., pressure) are transmitted to the nucleus to influence not only the renin expression but also chromatin structure and global gene expression, thus affecting the overall JG cell phenotype [66,69]. A myriad of additional systemic (salt intake, sympathetic nerve activity, hormones, etc.) and local (macula densa signaling, prostaglandins, NO, etc.) factors are also involved in the modulation of the renin production (for recent comprehensive reviews, see [66,70,71].

Plasticity is the central cellular mechanism regulating vascular renin production. Renin synthesis in the single cell appears to be constant [7,72,73]. Thus, the overall renin output is determined by the number of cells expressing renin, meaning that their number is variable and directly correlates to the renin production rate. The changes in the number of renin cells occur via reversible switching between an endocrine secretory and contractile smooth muscle phenotype. This phenomenon was originally termed “metaplastic transformation” but is currently clearly categorized as cellular plasticity [6,74,75,76]. Renin cell plasticity also has a pathophysiological role. Research conducted by our group and others has demonstrated that, following glomerular injury, renin-producing cells migrate into the glomerulus. Upon migration, these cells cease renin production and undergo differentiation to replace damaged glomerular cells [5,77,78,79,80]. Therefore, a single-nephron repair feedback mechanism exists that guides the migration of the renin-producing cells to the damaged area. At the same time, the migrating renin cell progeny remains in morphological contact with the JG cells in the corresponding afferent arteriole. Further studies are necessary to elucidate the precise matter of the signal(s) released by the injured glomerular cells, which initiate(s) the targeted movement and transformation of renin-producing cells in the juxtaglomerular part of the afferent arterioles.

Renin cell plasticity in the afferent arterioles of the mature kidney is pre-programmed during embryonic development. Thus, in the course of nephrogenesis, a subset of vascular smooth muscle cells originate from renin-producing precursors descending from Foxd1-positive stromal cells of the metanephric mesenchyme [5,81,82,83]. These vascular smooth muscle cells, particularly in the afferent arterioles of the adult kidney, retain a “memory” of their origin in a way that, upon conditions demanding increased renin production and activation of RAAS, they reversibly differentiate into renin-producing cells [6,76,84]. Many studies convincingly demonstrated the determinative role of cAMP as a second messenger for the formation and maintenance of the renin-producing cell identity [76,84,85,86,87,88,89].

Interestingly, cAMP/PKA signaling drives the AQP2 expression in principal cells and, thus, the water reabsorption in the collecting duct. Hence, it is not surprising that renin expression was also reported in principal cells. Albeit produced at a low level as compared to the juxtaglomerular cells, collecting duct renin seems to be expressed in more than half of the principal cells of the adult mouse kidney [7,90,91]. Developmental studies established that, compared to vascular renin, collecting duct renin is not decisive for proper nephrogenesis [83,92,93,94,95,96,97]. These data inferred that collecting duct renin would modulate the intrarenal effects of RAAS [98,99,100,101]. Indeed, studies demonstrated that collecting duct renin is important for potassium excretion and is involved in Ang II-dependent hypertension [92,102,103,104,105]. Ang II seems to be the major determinant of renin production in the collecting duct [101,106]. Therein, the cellular mechanism modulating the renin production was the cAMP/PKA/cAMP response element binding protein (CREB) transcription factor cascade as also established for the JG cells. A ubiquitously expressed (pro)renin receptor (PRR) has been mapped to the collecting duct. Originally described for its ability to bind and activate prorenin/renin, PRR is now known to have various RAAS-unrelated actions. PRR and its cleavage derivative, soluble PRR, exert complex physiological and pathophysiological effects in the collecting duct (recently reviewed in [100,107]).

### 3.3. Arguments for an Interplay between Cellular Plasticity and Renin Expression in the Collecting Duct

At first glance, the renin-producing cells and the collecting duct do not have many common features. Moreover, during nephrogenesis, they develop from different structures: the renin-producing cells from the metanephric mesenchyme [82] and the collecting duct from the ureteric bud [108]. However, going into more detail reveals some striking similarities: these cells produce renin, possess plasticity, and serve as master players in the control of blood pressure and the excretion of water and electrolytes. A link exists even during nephrogenesis: in the developing kidney, the collecting duct expresses renin so that the principal cells form the largest subpopulation of renin lineage cells in the adult kidney [91]. Together with cellular plasticity and the essential role in fluid homeostasis, the juxtaglomerular afferent arterioles and the collecting duct share striking functional characteristics. It is, therefore, plausible to assume that there may be a causal interplay in the collecting duct between plasticity on one side and renin expression on the other, as known for the JG cells in the afferent arterioles (Figure 1). There is a good amount of knowledge on cellular plasticity and renin production regarding the principal cells in the collecting duct. Although cellular plasticity and renin expression in the collecting duct have been generally regarded on a separate basis, several tenable mechanistic relations already exist, as summarized above. Nevertheless, the current knowledge provides rather indirect evidence for such a link and thus reveals a need for direct experimental confirmation in future studies.

## 4. Research Gaps/Open Questions/Perspectives

Studies focusing on (but not limited to) the following open questions would further address the putative interaction between cellular plasticity and renin expression in the collecting duct and surely fill research gaps to elucidate novel aspects of collecting duct function.

### 4.1. Does Renin Expression in the Collecting Duct Change When the Ratio among Principal and Different Subtypes of Intercalated Cells Is Altered (e.g., in Transgenic Animals, during Fluctuations of Acid–Base Balance, etc.)

Lithium has been described to affect the composition of the cells in the collecting duct [109]. In rats, lithium reduced the number of principal cells and led to an increase in intercalated cells. This effect is reversible after discontinuation of treatment. Lithium is used to treat patients with bipolar disorders. However, a side effect is that the majority of the patients develop an NDI, most probably reflecting the decreased number of principal cells [110]. The group of Mark Knepper analyzed changes in mRNA and protein expression in the micro-dissected cortical collecting duct and thick ascending limb of Henle (cTAL) from mice treated with lithium [111]. Unfortunately, they did not analyze if changes in cortical collecting duct cellular composition occurred.

Single-cell RNA sequencing data show that renin (*Ren1*) mRNA can be detected in several cell types of the mouse nephron, including the collecting duct (Figure 2) [112]. Interestingly, the expression of *Ren1* was highest in β-intercalated cells among all cell types examined. This was surprising because collecting duct renin is thought to be produced mainly by the principal cells. Therefore, the renin expression in β-intercalated cells should be further evaluated by independent approaches. It has already been reported that the intercalated cells are also at least partially of the renin lineage [91,113]. Furthermore, β-intercalated cells transdifferentiate under acidic conditions in response to homeostatic stress [51,52]. Assuming that renin mRNA expression is highest in β-intercalated cells among the collecting duct cell types [112], it might be hypothesized that these cells function as a progenitor cell niche in the collecting duct, with renin serving as a potential marker of cellular plasticity. Likewise, the renin-producing cells function as a reservoir of progenitor cells during nephrogenesis [81,82,83] and in the mature kidney after injury [77,78,79,80]. Therefore, it would be exciting to investigate whether the expression of typical β-intercalated cell genes or renin is modulated in β-intercalated cells during acidosis-induced transdifferentiation.

Another indication that β-intercalated cells could serve as a progenitor cell reservoir is the high expression of genes of the Wnt pathway (see Figure 2). Wnt signaling is crucial for embryonic development, cell division, tissue repair, and progenitor cell differentiation after injury [114,115]. Intriguingly, the Wnt pathway is activated during aging in renin-producing cells of the afferent arterioles, leading to increased renin expression and age-related renal fibrosis [116]. In addition, PRR, which is highly expressed in the collecting duct, serves both as a downstream target and co-activator of Wnt signaling [117]. This relationship suggests that PRR may play a role in modulating cellular plasticity in response to Wnt signaling within the collecting ducts. In line with this, Wnt and PRR antagonism emerge as promising strategies for treating chronic kidney disease [118,119]. However, the potential connection between PRR, renin, and cellular plasticity remains to be elucidated in future studies.
Figure 2Single-cell gene expression of Ren1, Aqp2, Slc4a1, Slc26a4, Slc4a9, Fzd5, Lrp5, Atp6ap2, and Ctnnb1 in segments and cells of the nephron, including the collecting duct. (**A**) Heat map illustrating mRNA expression of Ren1, Aqp2, Slc4a1, Slc26a4, Slc4a9, Fzd5, Lrp5, Atp6ap2, and Ctnnb1 along the nephron. The spatial distribution of the single-cell gene expression is indicated using the numbering at the top, with detailed nomenclature described in panel (**B**). Aqp2 serves as principal cells (#26) marker, Slc4a1 as intercalated cell type A (#25 and #27) marker, and Slc26a4 and Slc4a9 as intercalated cell type B (#24) markers. Ren1 exhibits the highest expression in #24. Fzd5, Lrp5, Atp6ap2 (PRR), and Ctnnb1, which are part of the Wnt pathway, most are also highly expressed in #24. (**B**) Schematic map depicting the anatomical locations of segments and cells in the nephron and the collecting duct, along with the corresponding nomenclature. LOH = loop of Henle, OM = outer medulla, IM = inner medulla, DST = distal straight tubule, CNT = connecting tubule, CCD = cortical collecting duct, OMCD = outer medullary collecting duct, IMCD = inner medullary collecting duct. Data from KidneyCellExplorer [112,120].
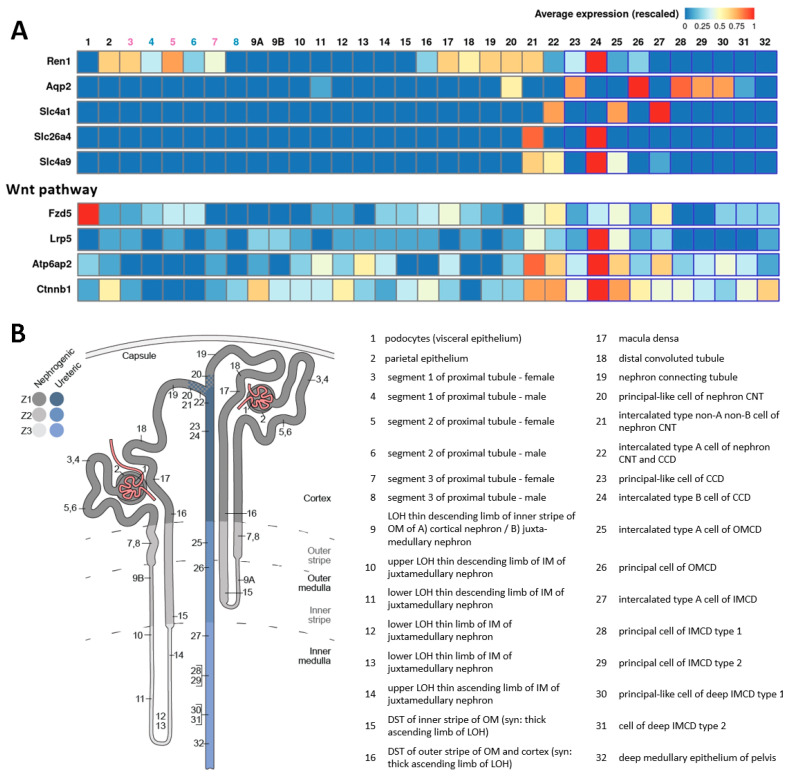



### 4.2. Does the Proportion of the Collecting Duct Cell Types Change When the Renin Expression in the Collecting Duct Is Altered (e.g., with Angiotensin II Treatment, in Collecting Duct-Specific Renin Knockout/Overexpressing Animals, during Different Developmental Stage, etc.)

Overexpression of renin, specifically in the principal cells of the collecting duct, is associated with hypertension in a transgenic mouse model [103]. The authors, however, did not analyze if this also has an impact on the ratio of intercalated cells vs. principal cells. The same group reported that a lack of renin in principal cells of the collecting duct attenuated Ang II-induced hypertension [92]. Also, in this study, the effect of renin deletion on the ratio of intercalated cells to principal cells was not evaluated. Another study showed that deletion of renin with Hoxb7-Cre had no impact on kidney morphology or circulating renin [83]. In both cases, it would be interesting to investigate if renin expression has an impact on the cell composition in the collecting duct (during both nephrogenesis and adult life).

The AE4 is expressed in both intercalated cell types; however, it is higher in β-intercalated cells (see Figure 2). Recently, a study showed that mice deficient in AE4 have a dysregulation in acid–base balance [42]. Whether this is associated with changes in the ratio between principal cells and intercalated cells was not analyzed. The effect of renin overexpression or deletion was only analyzed in principal cells. A study that used intercalated cell-specific overexpression or deletion of renin has not been performed so far.

Several studies showed that increased activation of RAAS could lead to chronic kidney disease [121]. However, a systematic analysis of whetherthis is also associated with changes in the composition of the intercalated cells/principal cells ratio, has not been performed. The biopsy samples derived from such studies are valuable resources that should be used to perform such an analysis.

### 4.3. Does the Cell Type Ratio or Renin Expression Change during Treatment with Diuretics

Diuretics like furosemide stimulate the expression of renin in the kidneys of salt-restricted mice and rats [122,123]. In addition, the activation of the V_2_-receptor by vasopressin induced increased renin expression [124]. Diuretic treatment leads to hypertrophy of the distal nephron and seems to foster cell proliferation in the collecting duct [125,126,127,128,129]. Yet, whether diuretics or antidiuretics affect the composition of the cells in the collecting duct in the long term has not been evaluated.

## 5. Conclusions

Renin expression in the renal collecting duct is well established. The same refers to the plasticity of the cells composing the collecting duct. Although studies analyzing the role of renin expression or the plasticity in the composition of the collecting duct cells have been performed, experiments primarily designed to reveal interactions between cellular plasticity and renin expression in the collecting duct are missing. This indicates the need for further investigations to analyze the potential causal interplay between these two features. Understanding this relationship could address unmet clinical needs, particularly in conditions where kidney function is compromised. For instance, clarifying the role of cellular plasticity and renin expression may improve our understanding of adaptive kidney responses or diseases such as nephrogenic diabetes insipidus, metabolic acidosis, and age-related renal fibrosis. Moreover, it could lead to optimizing the therapeutic use of diuretics and antidiuretics by minimizing side effects or enhancing efficacy.

Overall, this review sheds light on the multifaceted functions of the collecting duct and highlights the need for future research to elucidate the complex interactions between cellular plasticity and renin expression, potentially unveiling novel aspects of the collecting duct function and regulation.

## Figures and Tables

**Figure 1 ijms-25-09549-f001:**
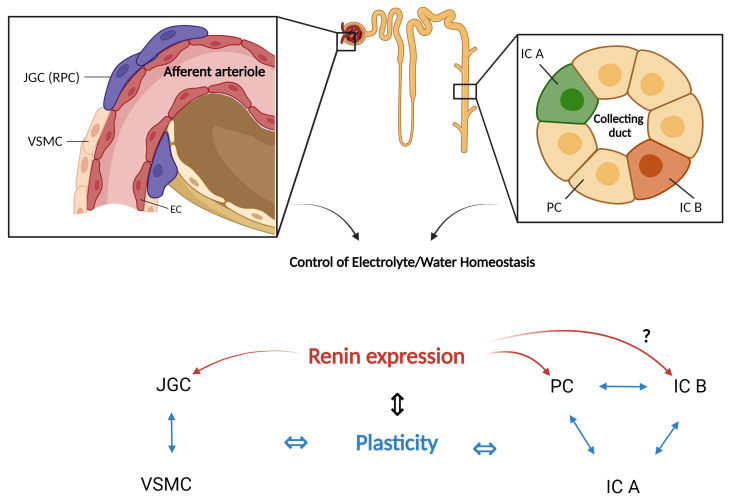
Juxtaglomerular afferent arterioles and the collecting duct share common characteristics, suggesting a potential interplay between plasticity and renin expression in the collecting duct, as known for the JG cells in the afferent arterioles. JGC = Juxtaglomerular cell, RPC = Renin-producing cell, EC = Endothelial cell, VSMC = Vascular smooth muscle cell, IC A = Intercalated cell type A, IC B = Intercalated cell type B, PC = Principal cell. See Figure 2 below regarding possible renin expression in intercalated cells type B. Created with BioRender.com.

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
