# Peer review of "A Possible Link between Cell Plasticity and Renin Expression in the Collecting Duct: A Narrative Review"

_ijms, 2024, doi:10.3390/ijms25179549_

Round 1

Reviewer 1 Report

Comments and Suggestions for Authors

The manuscript discussed the possible link between cell plasticity and renin expression in the collecting ducts. This is a very interesting topic, but there is still no direct evidence with very few studies have examined this question. This is just the author's speculation based on relevant literature.

1. What is the exact relationship between cell plasticity and renin expression in CDs? Mutual influence? It is total unclear.

2. As an important partner of renin, PRR, has similar expression patterns to renin in the CDs and highly expresses in the ICs with a relatively low expression in the PCs. Is there any correction between the cell plasticity, renin expression, and PRR expression in the CDs?

3. Regarding the single cell dataset, Ren1 gene expression is highest in the intercalated type B cells of the CD, which has not been validated and totally opposed to the current knowledge that renin expression has been validated in the principal cells of the CDs by ISH

4. Page 4 line 154, “The fact that intercalated cells can originate from principal cells has been shown……”, page 7 line 289, “……β-intercalated cells function as a pro- genitor cell niche in the collecting duct.”, line 292, “……β-intercalated cells could serve as a progenitor cell reservoir……”, contradictory.

5. Notch signaling in the collecting duct, which is important for cell plasticity, has not been described.

Author Response

Reviewer 1 Comment: The manuscript discussed the possible link between cell plasticity and renin expression in the collecting ducts. This is a very interesting topic, but there is still no direct evidence. This is just the author's speculation based on relevant literature.

Response: We gratefully acknowledge the concern of the Reviewer. We feel that it may arise from an unfortunate phrasing and presentation of the data. Therefore, we thoroughly edited the manuscript, to better focus on the existing evidence for a possible interplay between cellular plasticity and renin expression in the collecting duct. We also clearly admit that further studies should provide decisive experimental support to this topic.

Reviewer’s Comment 1: What is the exact relationship between cell plasticity and renin expression in CDs? Mutual influence? It is total unclear.

Response: We suggest that renin serves as a marker of progenitor cell subpopulation(s) in CD that are able to transdifferentiate during development and/or adult life. This is discussed in parts 1. Introduction and 4. Research Gaps / Open Questions / Perspectives of the revision. We also slightly edited the title as suggested by Reviewer 2 to stress that we provide our opinion on an underrated topic.

Reviewer’s Comment 2: As an important partner of renin, PRR, has similar expression patterns to renin in the CDs and highly expresses in the ICs with a relatively low expression in the PCs. Is there any correction between the cell plasticity, renin expression, and PRR expression in the CDs?

Response: We appreciate the Reviewer’s comment regarding the relationship between PRR, renin, and cellular plasticity in the collecting ducts. We acknowledge that while PRR and renin exhibit similar expression patterns and are both involved in cellular functions within the collecting ducts, the exact interplay between PRR, renin expression, and cellular plasticity remains to be elucidated. Likewise, we elaborated on this topic in part 4.1 of the revision.

Reviewer’s Comment 3: Regarding the single cell dataset, Ren1 gene expression is the highest in the intercalated type B cells of the CD, which has not been validated and totally opposed to the current knowledge that renin expression has been validated in the principal cells of the CDs by ISH

Response: We thank the Reviewer for this fair comment. We rephrased the manuscript and clearly stated that the high renin expression in intercalated type B cells requires further validation (part 4.1).

Reviewer’s Comment 4: Page 4 line 154, “The fact that intercalated cells can originate from principal cells has been shown…”, page 7 line 289, “ … β-intercalated cells function as a pro-genitor cell niche in the collecting duct.”, line 292, “ … β-intercalated cells could serve as a progenitor cell reservoir…”, contradictory.

Response: We gratefully acknowledge the concerns of the Reviewer. We feel that they may arise from an unfortunate phrasing and presentation of the data. Therefore, we revised the mentioned parts in the revision.

Reviewer’s Comment 5: Notch signalling in the collecting duct, which is important for cell plasticity, has not been described.

Response: We have addressed this important aspect by including a discussion on the role of Notch signaling in the collecting duct cellular plasticity (part 3.1).

Reviewer 2 Report

Comments and Suggestions for Authors

A Possible Link between Cell Plasticity and Renin Expression in Collecting Duct 

The authors present a well-written and thorough review of cell plasticity and renin expression. I have very few comments and in general see the manuscript as essentially suitable for publication.

Minor Comments

(1) Ideally the title of the manuscript should be accurate and descriptive, as readers may arrive at your paper via Google Scholar or PubMed and be choosing which papers to read based on titles alone. Perhaps

"A Possible Link between Cell Plasticity and Renin Expression in the Collecting Duct: a narrative review"

Would be better in highlighting that this is a narrative review? As for example used in https://doi.org/10.3390/biomedicines11112984

Note also that it is "the collecting duct", this should be corrected elsewhere as well, e.g. line 21, line 251.

(2) The manuscript is well written but gets very technical very quickly. It may be worth adding a new section 1. Introduction (the previously linked citation has an example). This could introduce the manuscript in a way suitable for a layperson with a broad understanding of science and medicine, and perhaps include a suitable visualisation to help the reader put the basics in context. This might be as simple as gathering information from other parts of the manuscript. 

(3) Is there any reason why Broeker et al (2022) has not been cited and discussed? It is at least as relevant as some of the other papers cited in this manuscript.

https://doi.org/10.1007/s00424-022-02694-8

(4) As a minor point, in the Introduction (once added) and Conclusions, it would be helpful to the reader to know if there are any unmet needs that the investigation of cell plasticity and renin expression would address. Highlighting a clinical need or a relevant disease or pathology can be helpful to the reader in understanding the significance of the work, such as the interplay of these issues with the aging process. This could even include better understanding of side effects of existing therapeutics, such as diuretics or antidiuretics. Addition of context as to why this research can produce benefits would add to the manuscript.

(5) A very minor point, but the last sentence of the Abstract does not read naturally. May I suggest instead:

"In this narrative review we discuss the link between these collecting duct features, which we see as a promising area for future research given recent findings."

Author Response

Reviewer 2 Comment:  The authors present a well-written and thorough review of cell plasticity and renin expression. I have very few comments and in general see the manuscript as essentially suitable for publication.

Response: We sincerely thank the Reviewer for the positive feedback and for considering our manuscript suitable for publication.

Reviewer’s Comment 1: Ideally the title of the manuscript should be accurate and descriptive, as readers may arrive at your paper via Google Scholar or PubMed and be choosing which papers to read based on titles alone. Perhaps

"A Possible Link between Cell Plasticity and Renin Expression in the Collecting Duct: a narrative review"

Would be better in highlighting that this is a narrative review? As for example used in https://doi.org/10.3390/biomedicines11112984 

Note also that it is "the collecting duct", this should be corrected elsewhere as well, e.g. line 21, line 251.

Response: We thank the Reviewer for this valuable suggestion. We agree that a more descriptive title would better reflect the nature of our work and assist readers in identifying the content of the manuscript. Therefore, we have followed the Reviewer’s recommendation and revised the title to "A Possible Link between Cell Plasticity and Renin Expression in the Collecting Duct: A Narrative Review." “the” was always inserted in front of “collecting duct”

Reviewer’s Comment 2: The manuscript is well written but gets very technical very quickly. It may be worth adding a new section 1. Introduction (the previously linked citation has an example). This could introduce the manuscript in a way suitable for a layperson with a broad understanding of science and medicine, and perhaps include a suitable visualisation to help the reader put the basics in context. This might be as simple as gathering information from other parts of the manuscript. 

Response: We gratefully acknowledge the Reviewer’s suggestion regarding the structure of the manuscript. To make the content more accessible to a broader audience, we have followed the recommendation and added a new Introduction section before the description of the functions of the collecting duct cells.

Reviewer’s Comment 3: Is there any reason why Broeker et al (2022) has not been cited and discussed? It is at least as relevant as some of the other papers cited in this manuscript.

https://doi.org/10.1007/s00424-022-02694-8 

Response: We appreciate the Reviewer bringing this to our attention. Accordingly, we have now cited this important work in the appropriate sections in the revised manuscript.

Reviewer’s Comment 4: As a minor point, in the Introduction (once added) and Conclusions, it would be helpful to the reader to know if there are any unmet needs that the investigation of cell plasticity and renin expression would address. Highlighting a clinical need or a relevant disease or pathology can be helpful to the reader in understanding the significance of the work, such as the interplay of these issues with the aging process. This could even include better understanding of side effects of existing therapeutics, such as diuretics or antidiuretics. Addition of context as to why this research can produce benefits would add to the manuscript.

Response: We appreciate the Reviewer's suggestion to enhance sections by emphasizing the clinical relevance and unmet needs that the investigation of cell plasticity and renin expression could address. We have revised the Conclusion to incorporate this perspective.

Reviewer’s Comment 5: A very minor point, but the last sentence of the Abstract does not read naturally. May I suggest instead:

"In this narrative review we discuss the link between these collecting duct features, which we see as a promising area for future research given recent findings."

Response: We appreciate the Reviewer’s attention to the clarity of our Abstract. In response to the suggestion, we have revised the last sentence to enhance its readability and natural flow. We believe this revision aligns with the Reviewer’s recommendation and improves the overall coherence of the Abstract.

Round 2

Reviewer 1 Report

Comments and Suggestions for Authors

I have no further comments.